# Ischemic Stroke Risk Associated with Mitochondrial Haplogroup F in the Asian Population

**DOI:** 10.3390/cells9081885

**Published:** 2020-08-11

**Authors:** Meng-Han Tsai, Chung-Wen Kuo, Tsu-Kung Lin, Chen-Jui Ho, Pei-Wen Wang, Jiin-Haur Chuang, Chia-Wei Liou

**Affiliations:** 1Department of Neurology, Kaohsiung Chang Gung Memorial Hospital and Chang Gung University College of Medicine, Kaohsiung 83301, Taiwan; menghan@cgmh.org.tw (M.-H.T.); tklin@adm.cgmh.org.tw (T.-K.L.); ultima1229@cgmh.org.tw (C.-J.H.); 2Center for Mitochondrial Research and Medicine, Kaohsiung Chang Gung Memorial Hospital and Chang Gung University College of Medicine, Kaohsiung 83301, Taiwan; bulakuo@icloud.com; 3Core Laboratory for Phenomics and Diagnostics, Kaohsiung Chang Gung Memorial Hospital and Chang Gung University College of Medicine, Kaohsiung 83301, Taiwan; 4Department of Metabolism, Kaohsiung Chang Gung Memorial Hospital and Chang Gung University College of Medicine, Kaohsiung 83301, Taiwan; wangpw@adm.cgmh.org.tw; 5Department of Pediatric Surgery, Kaohsiung Chang Gung Memorial Hospital and Chang Gung University College of Medicine, Kaohsiung 83301, Taiwan; jhchuang@adm.cgmh.org.tw

**Keywords:** mitochondrial DNA, ischemic stroke, cybrid model, hypoxia-ischemia, transcriptome, ANGPTL4

## Abstract

Mitochondrial dysfunction is involved in the pathogenesis of atherosclerosis, the primary risk factor for ischemic stroke. This study aims to explore the role of mitochondrial genomic variations in ischemic stroke, and to uncover the nuclear genes involved in this relationship. Eight hundred and thirty Taiwanese patients with a history of ischemic stroke and 966 normal controls were genotyped for their mitochondrial haplogroup (Mthapg). Cytoplasmic hybrid cells (cybrids) harboring different Mthapgs were used to observe functional differences under hypoxia-ischemia. RNA sequencing (RNASeq) was conducted to identify the particularly elevated mRNA. The patient study identified an association between Mthapg F1 and risk of ischemic stroke (OR 1.72:1.27–2.34, *p* = 0.001). The cellular study further demonstrated an impeded induction of hypoxic inducible factor 1α in the Mthapg F1 cybrid after hypoxia-ischemia. Additionally, the study demonstrated that Mthapg F cybrids were associated with an altered mitochondrial function, including decreased oxygen consumption, higher mitochondrial ROS production, and lower mitochondrial membrane potential. Mthapg F cybrids were also noted to be prone to inflammation, with increased expression of several inflammatory cytokines and elevated matrix metalloproteinase 9. The RNASeq identified significantly elevated expressions of angiopoietin-like 4 in Mthapg F1 cybrids after hypoxia-ischemia. Our study demonstrates an association between Mthapg F and susceptibility to ischemic stroke.

## 1. Introduction

Atherosclerosis is a progressive disorder and the most important risk factor for ischemic stroke [1]. Mitochondrial dysfunction has been identified to play a role in the pathogenesis of atherosclerosis through a variety of different mechanisms, such as increased production of reactive oxygen species (ROS), progressive respiratory chain dysfunction, and accumulation of mitochondrial DNA (mtDNA) damage [2]. Chronic overproduction of mitochondrial ROS leads to increased oxidation of low-density lipoprotein and dysfunction of endothelial cells that promote atherosclerosis. Impaired endothelial mitochondrial function also directly affects vascular cell growth and apoptosis [2]. Variations in mitochondrial genomic background, such as haplogroup, have been shown to exert different functional consequences [3], and be associated with diabetes and longevity [4,5].

The relationship between mtDNA haplogroup and ischemic stroke has been studied with inconclusive results. Mitochondrial DNA haplogroup K and H1 have been identified as protective factors in the European population (OR = 0.54 and OR = 0.61, respectively) [6,7], while mtDNA haplogroup D4b was found to be protective in Han Chinese patients (OR = 0.028) [8]. Recently, mtDNA haplogroup N9 was reported as an independent protective factor against neurological worsening in acute ischemic stroke patients [9]. On the contrary, haplogroup pre-HV/HV and U were found to be potential genetic risk factors for stroke in the European population (OR = 3.14 and OR = 2.87, respectively) [6]. Haplogroup A was associated with atherothrombotic cerebral infarction in Japanese female subjects, but not males [10]. Previous population studies demonstrated that the composition of mtDNA haplogroup varies among different ethnic groups [11,12]. Hence, it is possible that the risk of ischemic stroke is associated with different mtDNA haplogroups in different populations.

The aim of the present study was to investigate the association between mtDNA haplogroup and risk of ischemic stroke. We first recruited a cohort of Taiwanese patients to explore the relationship between mtDNA haplogroup and ischemic stroke. We then used cytoplasmic hybrid (cybrid) cells lines to demonstrate the discrepant functions within cells harboring different mtDNA haplogroups. By using RNASeq technology, we further identified nuclear genes potentially implicated in the association between a specific mtDNA haplogroup and risk of ischemic stroke.

## 2. Materials and Methods

### 2.1. Clinical Study

#### 2.1.1. Subject Recruitment for Association Study of Mitochondrial Haplogroups and Ischemic Stroke

Eight hundred and thirty ethnic Chinese ischemic stroke patients (538 males and 292 females) were recruited with an average age of 65.4 ± 11.0 years. The diagnosis of ischemic stroke was diagnosed clinically by experienced neurologists at the Department of Neurology, Linkou and Kaohsiung Chang-Gung Memorial Hospital based on clinical history and neuroimaging. Patients with hemorrhagic stroke, subarachnoid hemorrhage, and transient ischemic attack were excluded. A cohort of 966 ethnically, age- and sex-matched control subjects (570 male and 396 female) with a mean age of 64.2 ± 9.5 was recruited, who had received routine health examination and had been examined clinically without significant signs of neurological or cognitive impairment. Written informed consent was obtained from all participants, in accordance with protocols approved by the institutional review board at the Kaohsiung Chang-Gung Memorial Hospital (IRB number: 200501092B0, approved at 1 May 2005). The study was performed in accordance with the Declaration of Helsinki and its text revisions.

#### 2.1.2. Methods for Determination of Mitochondrial Haplogroup

Genomic DNA was extracted from venous blood by using a PUREGENE^®^ DNA Purification kit (Gentra, Minneapolis, MN, USA). Twenty-four pair primers (see Appendix A) and *ExTaq* DNA polymerase (Takara Bio Inc., Otsu, Japan) were used to amplify by multiplex polymerase chain reaction (PCR) and 94 probes (see Appendix A) were used to determine the mitochondrial haplogroup. Oligonucleotide probes were covalently bound to the carboxylated fluorescent microbeads using ethylene dichloride [5]. After hybridization, the amplicons were labeled with SA-PE using the Eppendorf Mastercycler gradient. Then, the reactions were measured by the Luminex100 flow cytometer [13]. By referencing the human mtSNP database provided in the Mitomap website, and the previously constructed phylogenetic trees for the Chinese population and the Japanese population, we selected 40 mtSNPs that define 15 major haplogroups (A, B, C, D, E, F, G, M7, M8, M9, M10, M11, M12, M13, N9) and their constitutive sub-haplogroups (B4, B5, D4, D5, F1, F2, F3, F4, M7a, M7b, M7c, M8a, N9a) in our population [11,14]. The mtSNPs for the corresponding haplogroups are shown in our previous publication [5].

#### 2.1.3. Subject Recruitment for Study of Serum ANGPTL4 Levels in Acute Ischemic Stroke

Blood samples of 86 stroke patients with an onset of neurological symptoms within 24 h and documentation of stroke by image studies as well as other evaluations, were collected for the study. Another group of blood samples from 61 subjects whom had participated in our health examination or out-patient service without ischemic stroke as shown by the image study, were collected as the control group. Written informed consent was obtained from all participants, in accordance with protocols approved by the institutional review board at the Kaohsiung Chang-Gung Memorial Hospital (IRB number: 201702300B0, approved at 1 March 2017). The study was conducted in accordance with the principles outlined in the Declaration of Helsinki.

#### 2.1.4. Methods for Determination of Serum ANGPTL4 Levels

ANGPTL4 protein levels were measured by the quantitative sandwich ELISA (catalog no. EA100609, OriGene, Rockville, MD, USA) according to the manufacturer’s instructions. Briefly, 100 μL of each serum sample was added into appropriate wells, incubated, then washed. After washing, biotinylated detection antibody, HRP-streptavidin solution, and ELISA colorimetric TMB reagent were sequentially added, then incubated at room temperature in the dark. Finally, a stop solution was added to each well, and the absorbance was measured at 450 nm on a spectrophotometer.

### 2.2. Cellular Study

#### 2.2.1. Cytoplasmic Hybrid (Cybrid) Cell Culture

MtDNA-depleted ρ0 cells were established by the treatment of human osteosarcoma 143B cells (BCRC 60439) (Bioresource Collection and Research Center, Taipei, Taiwan) with ethidium bromide for three months, and illustration of their mtDNA-less cellular status by RT-PCR has been previously described [15,16]. No misidentification or contamination of this cell line has also previously been confirmed after checking the NCBI database. Trans-mitochondrial cybrids were obtained by fusion of mtDNA-depleted ρ0 cells with platelets from donors harboring 20 of the major haplogroups found within the Asian population. These cybrids were cultured in Dulbecco’s Modified Eagle Medium (DMEM; Gibco BRL, Rockville, MD, USA) containing a 10% fetal bovine serum (Gibco, Carlsbad, CA, USA), 2 mM glutamine, 100 U/mL penicillin, and 100 μg/mL streptomycin (Gibco BRL, USA) in 5% CO2 at 37 °C. The cells were sub-cultured by treating with trypsin (0.05%)–EDTA (1:5000; Gibco BRL, Grand Island, NY, USA). The haplogroup of each cybrid was further confirmed by Sanger sequencing of mtDNA.

#### 2.2.2. Hypoxic-Ischemic Treatment

Cells were incubated in a PROOX model 110 chamber (Biospherix, Redfield, NY, USA) as previously described [17]. During incubation, a humidified environment at 37 °C, 95% N_2_, 5% CO_2_, and 1% O_2_ was maintained. Various hypoxic-ischemic time periods ranging from 0 and 24 h were used to determine the response. Glucose-free DMEM, 3-(4,5-dimethylthiazol-2-yl)-2,5diphenyl tetrazolium (MTT), 2′,7′-dichlorodihydrofluoresceindiacetate (H2DCF-DA), were used to mimic hypoxic-ischemic conditions.

#### 2.2.3. RNAseq Study and Analysis

In order to investigate the underlying biological mechanisms that cause the association between haplogroup F1 and ischemic stroke, we used NGS-based RNASeq to assess the alterations of mRNA expression after hypoxia-ischemia in different haplogroups. Total RNA was extracted by a Trizol^®^ reagent (Invitrogen, Carlsbad, CA, USA) according to the instruction manual. Purified RNA was quantified at OD260 nm by using a ND-1000 spectrophotometer (Nanodrop Technology, Wilmington, DE, USA) and qualified using a Bioanalyzer 2100 (Agilent Technology, Santa Clara, CA, USA) with an RNA 6000 labchip kit (Agilent Technologies, Santa Clara, CA, USA). Libraries of all samples were constructed using Agilent’s SureSelect Strand Specific RNA Library Preparation Kit for 75SE (single-end) and subsequently sequenced on an Illumina Solexa platform with the TruSeq SBS Kit. Raw reads were obtained and filtered using Trimmomatics to trim and remove the low-quality reads according to the quality score. Filtered reads were mapped and analyzed using TopHat and Cufflinks for gene expression estimation [18]. The gene expression level was calculated as FPKM (Fragments Per Kilobase of transcript per Million mapped reads). For differential expression analysis, CummeRbund was employed to perform statistical analyses of gene expression profiles. The reference genome and gene annotations were retrieved from the Ensemble database. Statistically significant expression changes between normoxia and hypoxia-ischemia in the same cell lines were estimated, and the false discovery rate adjusted *p*-value (*q*-value) was calculated. Only those differentially expressed genes (DEGs) triggered in response to hypoxia-ischemia with the log2 ratio ≥2 and *q*-value < 0.05 among four tested haplogroups were selected for the study.

#### 2.2.4. Western Blot for HIF-1α, IL-1β, IL-6, TNF-α, MMP-2, MMP-9, ANGPTL4

The cells were plated at a density of 1 × 10^6^ cells per well in 6-cm plates. After treatment in the hypoxic chamber, the cells were lysed with a buffer containing 150 mM NaCl, 50 mM HEPES pH 7, 1% Triton X-100, 10% glycerol, 1.5 mM MgCl_2_, 1 mM EGTA, and protease inhibitor, then harvested to isolate protein extract. To accurately compare protein expressions between individual cybrids, the total protein concentration in each cybrid cell was determined and 35 µg of protein was finally subjected to electrophoresis on 8–15% polyacrylamide gels. After the transfer, the polyvinylidene fluoride (PVDF) membrane (Millipore, Billerica, MA, USA) was blocked using a SuperBlock Blocking Buffer (Thermo Scientific, Waltham, MA, USA) for 0.5 h at room temperature, and then incubated overnight with primary antibodies at 4 °C. Following incubation of the secondary antibody with HRP, the signal imaging representing normoxia or hypoxia was analyzed with the same exposure time to the HRP substrate and captured by UVP BioSpectrum. Finally, signal intensity was quantified using the software ImageJ. After being normalized to a respective loading control such as actin, the fold change of protein expression in hypoxia was determined by comparing it to normal. The blotting was done with antibodies of HIF-1α (1:1000, Cell Signaling, Danvers, MA, USA), IL-1β (1:1000, GTX130021, GeneTex, Irvine, CA, USA), IL-6 (1:1000, GTX110527, GeneTex), TNF-α (1:500, GTX110520, GeneTex), MMP-2 (1:1000, GTX104577, GeneTex), MMP-9 (1:1000, GTX100458, GeneTex), anti-ANGPTL4 (1:1000, Signalway Antibody SAB, Pearland, TX, USA). The β-actin (1:1000, MAB1501, Millipore, Billerica, MA, USA) expression was used as the internal control.

#### 2.2.5. Energy Metabolism Assay for ATP, PDH, LDH, and Oxygen Consumption Rate

To determine cellular ATP levels, 7.5 × 10^4^ cells were trypsinized, washed, and resuspended in DPBS (Invitrogen, Carlsbad, CA, USA); supplemented with 2% FBS; and incubated in the presence of DMSO or oligomycin (Sigma, St. Louis, MO, USA) at 37 °C for 2 h. Cells were then collected to determine the ATP level (ATP Assay Kit no. K354-100, BioVision, Palo Alto, CA, USA) according to the manufacturers’ instructions. For the PDH and LDH assays, a Pyruvate Dehydrogenase Enzyme Activity Microplate Assay Kit (catalog no. K679-100, BioVision, Milpitas, CA, USA) and Lactate Dehydrogenase (LDH) Assay Kit (catalog no. ab102526, Abcam, Cambridge, MA, USA) were used following the manufacturer’s protocol. Oxygen consumption in cells and mitochondria were determined using the Seahorse XF24 Extracellular Flux Analyzer (Seahorse Bioscience Inc., Chicopee, MA, USA). To facilitate comparison between experiments, data were presented as OCR in pmol/min/1 × 10^5^ cells and ECAR in mpH/min/1 × 10^5^ cells. Cells were seeded in triplicate at a density of 1 × 10^5^ cells per well in Seahorse cell culture 24-well plates and cultured with a serum-free medium. After 24 h of incubation, under normoxic or hypoxic (1% oxygen) conditions, the cells were washed out using 1 mL of the Seahorse medium (DMEM without sodium bicarbonate), then 675 μL of the Seahorse medium was added to each well. Basal OCR was measured four times and plotted as a function of cells under basal conditions followed by the sequential addition of 1 μM oligomycin, 250 nM carbonyl cyanide p-trifluoromethoxyphenylhydrazone (FCCP), and 2 μM rotenone, as indicated. At the end of the recording period, cells were collected and cell viability was determined using a trypan blue exclusion assay. OCR values were calculated after normalizing with the cell number.

#### 2.2.6. Flow Cytometry Analysis for ROS, Calcium, and Membrane Potential

The levels of mitochondrial superoxide (O_2_^−^) was detected using a 1 µM MitoSOX™ Red mitochondrial superoxide indicator (Molecular probe, Invitrogen, Thermo Fisher Scientific, Inc., Waltham, MA, USA) in Hank’s balanced salt solution for 10 min at 37 °C. The intracellular ROS products were evaluated by measuring the levels of hydrogen peroxide (H_2_O_2_) using 2′,7′-dichlorofluorescin diacetate (DCFH-DA; Sigma, St. Louis, MO, USA) for 30 min at 37 °C. the mitochondrial calcium level was measured by using 0.5 μg/mL Rhod2-AM staining (molecular probe) for 30 min at 37 °C. The intracellular calcium indicator was determined by using 10 μM Fura-2 AM (molecular probe) for 1 h at 37 °C. Rhodamine 123 is a mitochondria-specific fluorescent dye and a protocol with 0.5 μg treatment for 30 min at 37 °C was used to measure mitochondrial membrane potential under the hypoxic-ischemic condition. Cells were plated at a concentration of 5 × 10^5^ cells per well in 6-well plates. After treatment in the hypoxic chamber, an individual analysis was conducted with a specific dye. After incubation, the cells were harvested and washed twice with PBS and finally added into 1 mL PBS. Finally, FACS Calibur flow cytometer (BD Biosciences, San Jose, CA, USA) was used to detect the fluorescence intensity of intracellular and mitochondria. The data was analyzed using the flow cytometry analysis software WinMDI 2.8.

### 2.3. Statistical Analysis

Statistical analysis was performed using the Statistical Package for the Social Sciences software (SPSS, IBM, Armonk, NY, USA). We performed multivariate logistic regression analysis to adjust for risk factors, with ischemic stroke as a dependent variable and independent variables including age, sex, major risk factors, and mtDNA haplogroups. Results were presented as the odds ratio (OR) and 95% CI. For comparison of data regarding the rate differences of mtDNA variants between various haplogroups, the Chi-square test was conducted. Some haplogroups with a limited number of cases were allocated into groups of “others in N or M” for comparison according to their related macro-haplogroup in the phylogenetic tree. Bonferroni’s correction was used to correct for multiple comparisons of mtDNA haplogroups. Since we examined 16 haplogroups (A, B4, B5, C, D4, D5, E, F1, F2, G, M7b, M7c, M8, N9, others in N, others in M), we divided 0.05 by 15 to arrive at 0.0031. Thus, a *p-*value of < 0.0031 was considered statistically significant. For the ANGPTL4 serum level and cellular model studies, statistical analysis was performed using an independent-samples T test or one-way ANOVA computation. A *p*-value < 0.05 was considered statistically significant.

## 3. Results

### 3.1. Clinical Genetic Study

#### Notable Associations between Specific mtDNA Haplogroups and Ischemic Stroke

From a cohort of 830 patients having suffered ischemic stroke, we investigated the association between mtDNA haplogroup and occurrence of ischemic stroke. Multiple logistic regression analysis revealed that haplogroup F was significantly associated with an increased ischemic stroke risk; whereas haplogroup B was identified as significantly associated with a decreased risk of ischemic stroke (OR 0.77, 95% CI = 0.61–0.97, *p* = 0.024). However, after Bonferroni correction, haplogroup B failed to reach significance, and only haplogroup F remained significant (OR 1.44, 95% CI = 1.14–1.82, *p* = 0.002). Further sub-group analysis identified haplogroup F1 as significantly associated with an increased risk of ischemic stroke (OR 1.72, 95% CI = 1.27–2.34, *p* = 0.001); meanwhile, although an association was also identified with haplogroup F2, this association was rendered insignificant after Bonferroni correction (OR 1.68, 95% CI = 1.13–2.48, *p* = 0.01) (Table 1).

### 3.2. Cellular Function and Protein Expression Study

#### 3.2.1. Inferior HIF-1α Response to Hypoxia-Ischemia in Specific Haplogroup Cybrids

Hypoxia-inducible factor 1 alpha (HIF-1α) is a key factor regulating cellular adaptive and survival responses to hypoxia-ischemia, we therefore evaluated the protein expressions of HIF-1α under hypoxia-ischemia [19]. Marked HIF-1α level increases were observed in all cybrids harboring different haplogroups under hypoxic-ischemic stress. However, stroke-susceptible haplogroup F1 exhibited the least elevated HIF-1α level; whereas haplogroup B exhibited the most elevated level in response to hypoxia-ischemia (Figure 1). To confirm that the elevated protein signal was not caused by discrepancies of exposure time, we also conducted all samples on the same page and achieved the similar result (Appendix A).

#### 3.2.2. Notable Alterations of Metabolic Profiles in Specific Haplogroup Cybrids after Hypoxia-Ischemia

We investigated whether different mitochondrial genomic alterations influence mitochondrial functional profiles under hypoxic-ischemic conditions. The colorimetric assay illustrates that hypoxia-ischemia diminished ATP production in all haplogroups (Figure 2A). As pyruvate dehydrogenase (PDH) and lactate dehydrogenase (LDH) are key enzymes for aerobic/anaerobic metabolism, we investigated their activity in different haplogroups in response to hypoxia-ischemia. The PDH activity was significantly reduced from normoxia to hypoxia-ischemia in cybrids F1 and F2 (Figure 2B). However, although a tendency of elevated LDH levels were observed in all cybrids after the hypoxic-ischemic treatment, these changes lacked significance (Figure 2C). The oxygen consumption rate (OCR) in the absence or presence of oligomycin, FCCP, and rotenone, under normoxia (Figure 2D) and hypoxia (Figure 2E), demonstrated that the basal, coupled, maximal, and spare respiration of cybrids F1 and F2 were substantially lower than that of cybrids B4 and B5 (Figure 2F–H). Whereas, the extracellular acidification rate (ECAR) exhibited no significant difference between cybrids.

#### 3.2.3. Notable Alterations of Mitochondrial Function Profiles in Specific Haplogroup Cybrids after Hypoxia-Ischemia

As hypoxia is known to increase mitochondrial reactive oxygen species (ROS) generation, we investigated both the cytosolic and mitochondrial ROS with DCFH and MitoSOX red. Hypoxia increased mitochondrial and cytosolic ROS generation in both cybrids B and F. However, while cybrid B exhibited a notably higher level of cytosolic ROS than cybrid F1 under hypoxic-ischemic conditions (Appendix A), the elevation in mitochondrial ROS was less pronounced in ischemic-susceptible mitochondrial haplogroup F1 than in cybrid B4 (Figure 3A). In addition, mitochondrial membrane potential Δ*ψ*_m,_ which represents mitochondrial function, was significantly decreased in haplogroup F1 compared to haplogroup B4 after hypoxia-ischemia (Figure 3B).

#### 3.2.4. Elevated Intracellular Calcium Levels in Specific Haplogroup Cybrids after Hypoxia-Ischemia

Since calcium has been shown to be involved in the regulation of mitochondrial function, the introduction of large amounts of mitochondrial calcium under extended hypoxia-ischemia, and the subsequent loss of intracellular calcium homeostasis, could be critical to the triggering of cellular necrosis [20]. We therefore investigated the distribution of intracellular calcium in response to hypoxia-ischemia in different haplogroup cybrids. We found that hypoxia-ischemia induced increases of cytosolic Ca^2+^ concentrations, particularly significant in the F2 cybrids (Figure 3C). Furthermore, markedly elevated levels of intra-mitochondrial calcium were observed in the F1 and F2 cybrids, as compared to haplogroup B4 cybrids (Figure 3D).

#### 3.2.5. Discrepancies in Levels of IL-1β, IL6, TNF-α, and MMP-2 among Different Haplogroup Cybrids after Hypoxia-Ischemia

We investigated the different responses to hypoxic-ischemic stress on the regulation of inflammation and the extracellular matrix. The levels of inflammatory cytokines, including interleukin-1β (IL-1β) and interleukine-6 (IL-6) were elevated after hypoxia-ischemia in all cybrids but significantly higher in haplogroup F cybrids, while the level of tumor necrosis factor-alpha (TNF-α) was elevated in F2 cybrids (Figure 4A–C). Additionally, the elevation of matrix metalloproteinase 9 (MMP-9) after hypoxia-ischemia was significant (Figure 4D), whereas induction of MMP-2 was relatively limited in haplogroup F cybrids compared to haplogroup B cybrids (Appendix A).

#### 3.2.6. RNAseq Study Identified Increased Expression of ANGPTL4 in Haplogroup F1 Cybrid Cells in Response to Hypoxia-Ischemia

The initial NGS-based RNASeq analysis revealed 16 DEGs in haplogroup F1, 27 DEGs in haplogroup F2, 9 DEGs in B4, and 11 DEGs in B5. After further comparison of the different haplogroups, we found that only two genes were specifically expressed in F1 but not the other three haplogroups: *ANGPTL4* and *FAP* (Figure 5). The *FAP* gene was downregulated; whereas ANGPTL4 showed increased expression (log2 ratio = 5.61). *FAP* encodes fibroblast activation protein alpha, which has been reported to be associated with inflammation and stroke [21]; meanwhile, *ANGPTL4* encodes angiopoietin-like 4, a protein expressed in vascular endothelial cells, which may have an influence on vascular atherosclerosis and ischemic stroke. Protein expressions of ANGPTL4 in different cybrids after hypoxia-ischemia were then further analyzed. Consistent with the RNASeq analysis, the expression of ANGPTL4 had a marked increase in haplogroup F cybrids compared to haplogroup B cybrids (Figure 6C). However, the protein study of FAP did not demonstrate a consistent result in Western blotting. To further determine the relationship between HIF-1α and ANGPTL4, we used siRNA to study the expressions of HIF-1α and ANGPTL4 after hypoxia-ischemia. While the knockdown of HIF-1α suppressed expression of HIF-1α (Figure 6A,B), it did not significantly affect the expression of ANGPLT4 after hypoxia-ischemia (Figure 6D), suggesting that ANGPLT4 expression is independent from HIF-1α induction.

#### 3.2.7. Elevation of ANGPTL4 Serum Levels in Acute Ischemic Stroke Patients

It has been reported that ANGPTL4 serum levels may increase during a stroke event [22]. Therefore, to test the potential use of serum ANGPTL4 levels as a biomarker for correlative diagnosis of acute ischemic stroke, we analyzed the serum levels in blood samples taken from a small cohort of patients within one day of an ischemic stroke event. A higher average level of ANGPTL4 was found in the acute ischemic stroke patient cohort (4941 ± 2663 pg/mL) compared with the control (3313 ± 1511 pg/mL). The difference remained significant after adjustment for age, sex, and other major risk factors for ischemic stroke (4941 ± 2663 vs. 3313 ± 1511, *p* = 0.001).

## 4. Discussion

In this study, we identified specific mtDNA variants, which define the mitochondrial haplogroup F as susceptible to ischemic stroke. Further sub-group analysis suggests that this susceptibility is associated with sub-haplogroup F1 in particular. Previous studies have demonstrated that mtDNA haplogroups pre-HV/HV, and U are risk factors for ischemic stroke in the European population; meanwhile, mtDNA haplogroup A has been associated with atherothrombotic cerebral infarction in Japanese females [6,10]. Although the underlying pathogenesis between ischemic stroke and different mtDNA haplogroups remains unclear, the non-synonymous variant G13928A in mtDNA haplogroup F and additional non-synonymous variants found in mtDNA haplogroup F1, could affect oxidative stress due to alterations in amino acid and protein characteristics [23,24] (see Appendix A). Haplogroup F in particular is the second-largest mtDNA haplogroup harbored by the ethnic Chinese population of southern China, and harbored by an estimated 600 million people in the East Asia region [25]. Our findings are therefore particularly relevant to the development of clinical applications to identify the risk of ischemic stroke among Asian population groups.

The cellular model study revealed that the induction of HIF-1α after hypoxia-ischemia is impeded in cybrids harboring stroke-susceptible F1 mtDNA haplogroups. Of note, HIF-1α has been reported to have protective effects against a stroke-related injury during hypoxic conditions. Through the promotion of angiogenesis, erythropoietin expression, and rapid adaptation to hypoxic stress via triggering of the anaerobic metabolism, HIF-1α can enhance cellular endurance to hypoxia-ischemia [19,26]. The relatively reduced induction of HIF-1α in haplogroup F1 cybrids after hypoxia-ischemia further supports the clinical findings of a vulnerability to ischemic stroke in patients harboring mtDNA haplogroup F1.

Our study also demonstrated that various mitochondrial functions of haplogroups F1 and F2 are notably decreased during hypoxia-ischemia. While all cybrids were found to have lower ATP availability, a significant reduction in PDH activity and relatively inferior response to hypoxic-ischemic scenario of LDH, indicates that haplogroup F cannot efficiently adapt to the anaerobic pyruvate-lactate cycle in a low oxygen condition. This is further supported by the lower oxygen consumption rate of F cybrids in the basal, coupled, maximal, and spare respiration under normoxic and hypoxic conditions. The increased ratio of mitochondrial ROS to cytosolic ROS, and reduced mitochondrial membrane potential, further suggest an altered mitochondrial function under hypoxic-ischemic stress. Taken together, haplogroup F has impaired protective responses to a low-oxygen condition, hence is vulnerable to the pathogenesis of ischemic stroke.

By using mRNA profiling and cybrids harboring the same nuclear gene but different mtDNA haplogroups, we identified nuclear genes possibly implicated in the association between specific mtDNA haplogroups and a risk of ischemic stroke. The RNASeq analysis revealed that expression of ANGPTL4, previously found to be associated with ischemic stroke [22,27], was particularly elevated in mtDNA haplogroup F1 after hypoxic-ischemic stress. In previous studies, the ANGPTL4 activity has been associated with endothelial cell integrity, inflammation, oxidative stress, and lipid metabolism [28]. In addition, ANGPTL4 is known to regulate cellular energy homeostasis and ROS, and has been implicated in aspects of the pathogenesis of atherosclerosis, including endothelial dysfunction, oxidation of LDL, and reduced nitric oxide [29]. Therefore, the increased risk of stroke in patients harboring haplogroup F1 could be attributed to the significantly elevated expression of ANGPLT4. This finding warrants further investigation in future studies. Further investigation of the relationship between HIF-1α and ANGPTL4 indicated that ANGPLT4 expression is independent from HIF-1α induction. Although the role of ANGPTL4 in ischemic stroke remains inconclusive [30], recent studies have identified elevated protein expressions of ANGPTL4 in stroke patients [27]. Meanwhile, a separate study noted that genetic polymorphisms in ANGPTL4 are associated with an increased risk of large artery atherosclerotic stroke [22]. Of note, our patient group study identified significantly elevated levels of ANGPTL4 in the acute ischemic stroke cohort, indicating the clinical potential of utilizing the ANGPTL4 level as a biomarker for the identification of acute ischemic stroke [30].

We also found that the expressions of inflammatory cytokines IL1, IL-6, and TNF-α, were all particularly elevated in haplogroups F1 and F2 cybrids after hypoxia-ischemia, concurrent with the increased expression of ANGPTL4, a key modulator of both inflammatory response and vascular permeability [27]. Of note, inflammation has been recognized as an important contributor to the formation of atherosclerosis [31]. The elevated expressions of ANGPTL4 and inflammatory cytokines suggest that the risk of ischemic stroke in haplogroup F may be partially due to its relatively enhanced inflammatory response during hypoxic-ischemic conditions, potentially facilitating the process of atherosclerosis.

Activation of matrix metalloproteinases (MMPs) in the brain after ischemia has been reported, including increased expression of constitutive enzymes (MMP-2) and inducible enzymes (MMP-9) [32]. These MMPs are known to play multiple roles in ischemic injury, participating in the early stages of the injury process and contributing to recovery during the later stages [33]. In our study, we found that MMP-9 expression is significantly induced, whereas MMP-2 induction is relatively limited in stroke-susceptible haplogroup F cybrids in comparison with B cybrids. Previous studies have reported that MMPs, particularly MMP-9, were detected in the acute phase after ischemic stroke, coinciding with the opening of the blood–brain barrier, and partially contributing to the extent of the infarction [34]. In contrast, the MMP-2 protein level has been observed to increase several days after ischemic stroke, when barrier leakage is presumably restored, and formation of glial scarring begins [32,34]. The results herein indicate an elevated vulnerability for those harboring mtDNA haplogroup F to a risk of ischemic stroke.

Mitochondria detect and define cytosolic calcium levels by importing and subsequently releasing calcium during periods of physiological and pathological calcium elevations. Extended periods of elevated mitochondrial matrix calcium concentration have been suggested as a primary cause of neuronal death during hypoxia-ischemia [35]. In the present study, we observed discrepant expressions of calcium homeostasis after hypoxia-ischemia in cybrids harboring different mitochondrial haplogroups. Consistent with previous studies, cytosolic calcium levels increased in all cybrids. Furthermore, we demonstrated that mitochondrial calcium levels increased in all cybrids, but the increase was markedly higher in stroke-susceptible haplogroup F. As dysregulation of the mitochondrial calcium level has previously been implicated in cellular injury, our findings suggest that haplogroup F cybrids appear to have a relatively inferior calcium-dependent adaptation response after hypoxia-ischemia, which may contribute to the pathogenesis of ischemic stroke.

The primary limitation to the cellular study is that the cybrid model we used effectively eliminated the influence of the nuclear genome, we therefore did not investigate the impact of different nuclear genomes on the risk of ischemic stroke. It is possible that a nuclear genomic variation inherited together with the mitochondrial haplogroup could contribute to a stroke risk. With regards to the clinical study, due to limited patient numbers, we did not analyze different subgroups of ischemic stroke patients. Future clinical study of a larger patient cohort including subgroup analysis of ischemic stroke types is recommended.

## 5. Conclusions

In conclusion, we identified mitochondrial haplogroup F as a risk factor for ischemic stroke. Functional assays demonstrated that cybrids harboring haplogroup F have an increased expression of ANGPLT4 and decreased mitochondrial function, which may lead to the accumulation of ROS and calcium in mitochondria and increased inflammatory response. The findings of this study indicate that cells harboring haplogroup F have an inferior response to hypoxic-ischemic conditions, possibly predisposing certain subjects to the risk of ischemic stroke.

## Figures and Tables

**Figure 1 cells-09-01885-f001:**
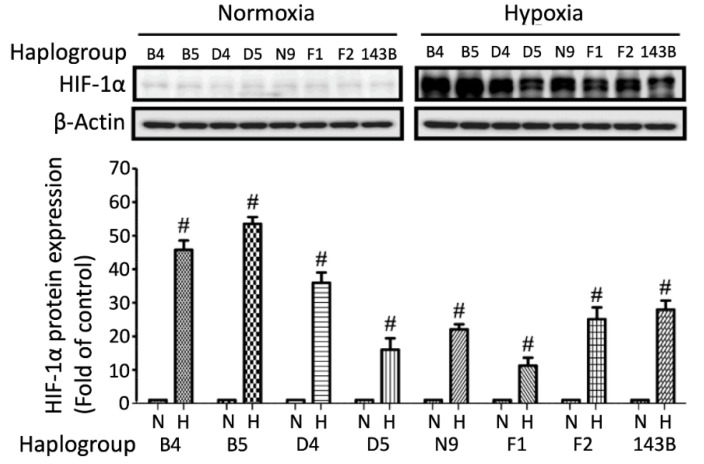
HIF-1α expressions in different cybrids under hypoxia-ischemia. Comparison of HIF-1α expression under hypoxic-ischemic conditions in cybrids harboring common mitochondrial haplogroups found in the ethnic Chinese population (B4, B5, D4, D5, F1, F2, N9). The 143B cybrid was used to represent the Caucasian population. Actin was used as a loading control. Data represent the mean ± SD of at least three independent experiments (*^#^ p* < 0.005). N was used to represent normoxic and H to represent the hypoxic condition for 24 h.

**Figure 2 cells-09-01885-f002:**
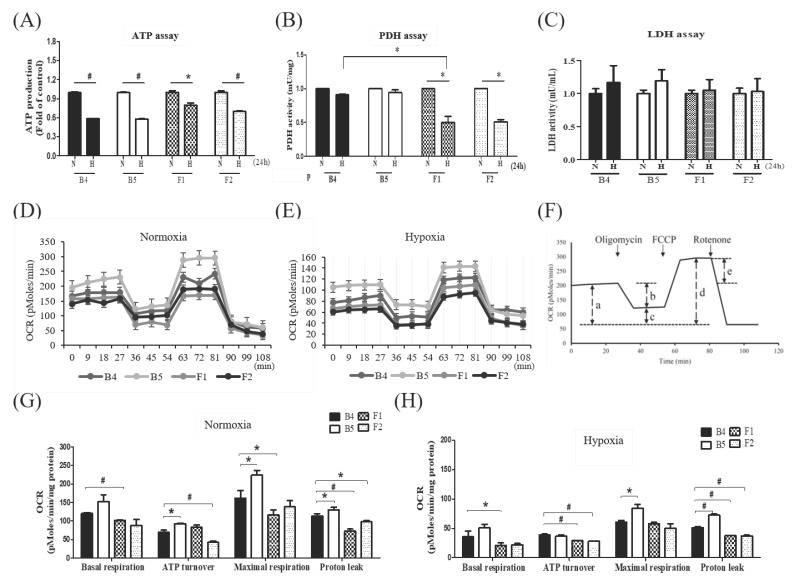
Metabolic profile of different cybrids after hypoxia-ischemia. Comparison of ATP production (**A**), pyruvate dehydrogenase (PDH) activity (**B**), lactate dehydrogenase (LDH) activity (**C**) under hypoxic-ischemic condition in cybrid cells harboring stroke-susceptible haplogroup F and stroke-resistant haplogroup B. Measurements of oxygen consumption rate (OCR) under normoxic (**D**) and hypoxic (**E**) conditions, and comparison of the fundamental OCR parameters under normoxic (**G**) and hypoxic (**H**) conditions, in cybrid cells harboring stroke-susceptible haplogroup F and stroke-resistant haplogroup B. (**F**) represents the fundamental parameters of mitochondrial OCR including basal respiration (a), coupled respiration (b), uncouple respiration (c), maximal respiration (d), and spare respiratory capacity (e). Mitochondrial ATP levels, PDH and LDH activity were determined by the colorimetric assay kit. Data represent the mean ±SD of at least three independent experiments (* *p* < 0.05; ^#^
*p* < 0.005). N was used to represent normoxic and H to represent hypoxic condition. OCR examined by the Seahorse XF24 analyzer in the absence or presence of oligomycin (1 μM), FCCP (250 nM), and rotenone (1 μM) under normoxic or hypoxic conditions, data obtained by counting 1 × 10^5^ cells of cybrids.

**Figure 3 cells-09-01885-f003:**
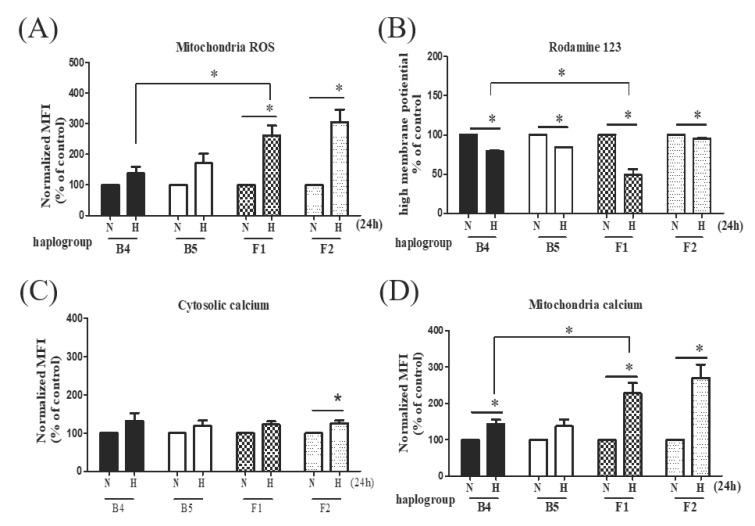
Mitochondrial profile of different cybrids after hypoxia-ischemia. Comparison of mitochondrial ROS (**A**), mitochondrial transmembrane potential (**B**), cytosolic calcium levels (**C**), and distribution of mitochondrial calcium (**D**) under hypoxic-ischemic conditions in cybrid cells harboring stroke-susceptible haplogroup F and stroke-resistant haplogroup B. MitoSOX-based flow cytometry was used for detecting mitochondrial ROS. The fluorescence of mitochondrial transmembrane potential was measured by Rhodamine 123. The mobilization of distinct pools of calcium between cytosolic and mitochondria, stained with Fura2-AM and Rhod-2 AM, respectively, were measured by flow cytometry analysis. Data represent the mean ± SD of at least three independent experiments (** p* < 0.05). N was used to represent normoxic and H to represent the hypoxic condition.

**Figure 4 cells-09-01885-f004:**
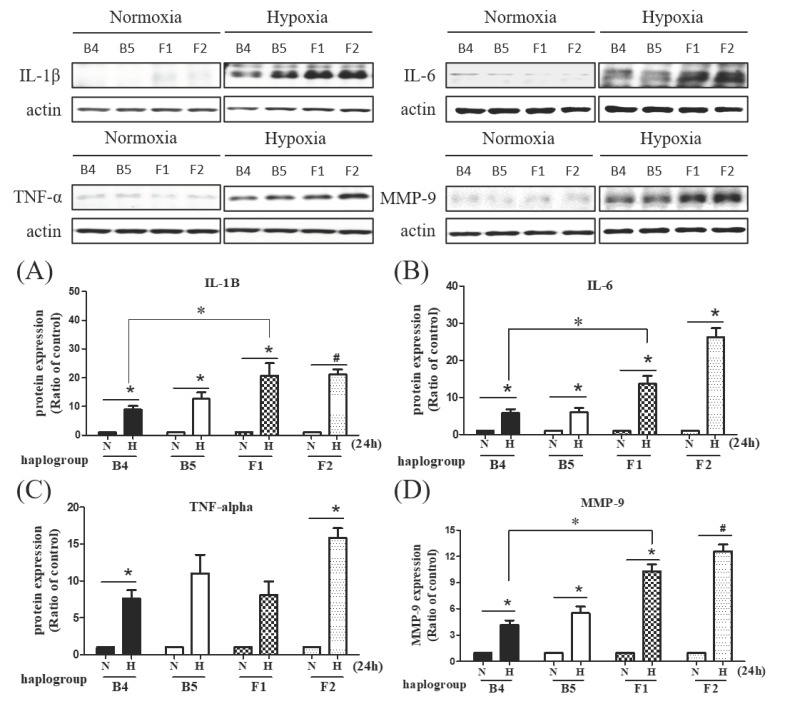
The expression of inflammatory markers and extracellular matrix in different cybrids after hypoxia-ischemia. Comparisons of inflammatory cytokines in cybrid cells harboring stroke-susceptible haplogroup F and resistant haplogroup B. Western blot analysis shows hypoxia induced protein expression of IL-1β (**A**), IL-6 (**B**), TNF-α (**C**), and MMP-9 (**D**). Actin was used as loading controls. Data represent the mean ± SD of at least three independent experiments (** p* < 0.05; ^#^
*p* < 0.005). N was used to represent normoxic and H to represent the hypoxic condition.

**Figure 5 cells-09-01885-f005:**
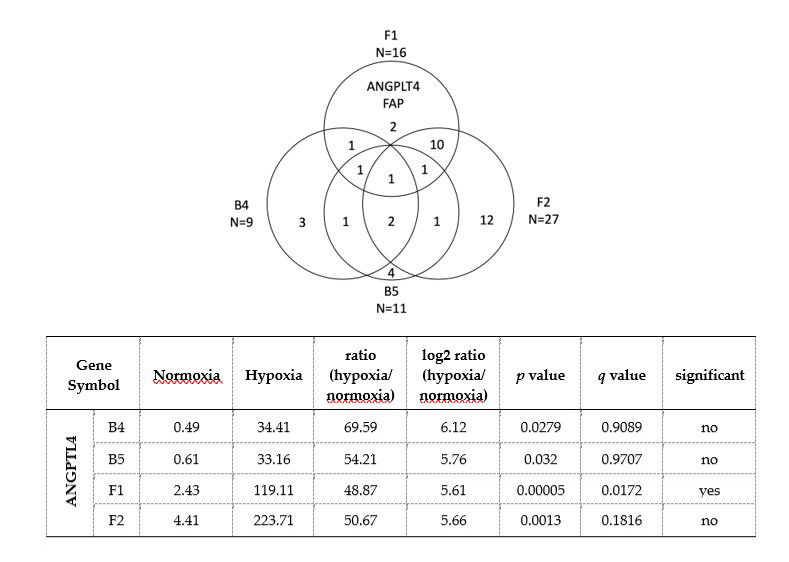
Gene expression profiling of F1 and F2 cybrids compared to B4 and B5 cybrids using the RNASeq technology. The Venn diagram illustrates the number of differentially expressed genes (DEGs) between hypoxic-ischemic for 24 h and normoxic conditions in cybrids harboring haplogroups F1, F2, B4, and B5. The two genes specific to stroke-susceptible haplogroup F1 were ANGPLT4 and FAP. FAP: Fibroblast activation protein alpha; ANGPLT4: Angiopoietin like 4; FPKM: Fragments Per Kilobase of transcript per Million mapped reads; *q*-value, false discovery rate (FDR) adjusted *p*-value.

**Figure 6 cells-09-01885-f006:**
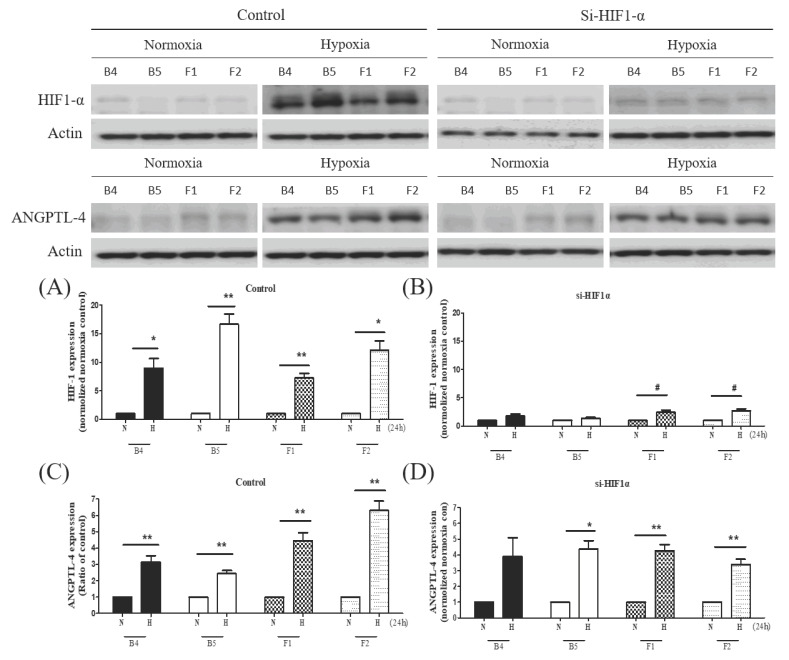
ANGPTL4 expression in different cybrids under hypoxia-ischemia and the relationship with HIF-1α. Comparison of the knockdown effect of HIF-1α siRNA on expressions of HIF-1α and ANGPTL4 in cybrid cells harboring stroke-susceptible haplogroup F and resistant haplogroup B. Baseline expression of HIF-1α (**A**) and ANGPTL4 (**C**) and effects of HIF-1α siRNA on expressions of HIF-1α (**B**) and ANGPTL4 (**D**) before and after hypoxia-ischemia, as assessed by Western blot analysis. Actin was used as a loading control. Data represent the mean ±SD of at least three independent experiments (* *p* < 0.05; *^#^ p* < 0.005). N was used to represent normoxic and H to represent the hypoxic condition.

**Table 1 cells-09-01885-t001:** Multivariate logistic regression analysis of mitochondrial haplogroups associated with ischemic stroke.

Haplogroup	Stroke (*n* = 830)	Control (*n* = 966)	Total (*n* = 1796)	Multivariate	
	% (*n*)	% (*n*)	% (*n*)	Odds Ratio (95% CI)	*p*
Major haplogroup					
A	5.1 (42)	4.4 (42)	4.6 (82)	1.16 (0.74–1.81)	0.529
B	20.2 (168)	24.9 (240)	22.7 (408)	0.77 (0.61–0.97)	0.024
C	2.5 (21)	2.6 (25)	2.6 (46)	0.93 (0.51–1.69)	0.806
D	17.6 (146)	18.4 (178)	18.0 (324)	0.94 (0.73–1.20)	0.599
E	2.0 (17)	1.9 (18)	2.0 (35)	1.31 (0.66–2.60)	0.441
F	24.8 (206)	18.6 (180)	21.5 (386)	1.44 (1.14–1.82)	0.002 *
G	2.5 (21)	2.5 (24)	2.5 (45)	1.10 (0.60–2.03)	0.751
M7	11.8 (98)	12.6 (122)	12.3 (220)	0.91 (0.68–1.22)	0.544
M8	4.7 (39)	3.9 (38)	4.4 (79)	1.27 (0.80–2.03)	0.312
N9	3.5 (29)	2.8 (27)	3.1 (56)	1.28 (0.77–2.11)	0.289
Others N	1.2 (10)	1.8 (17)	1.5 (27)	0.74 (0.33–1.64)	0.460
Others M	4.0 (33)	5.7 (55)	4.9 (88)	0.61 (0.39–0.96)	0.031
Sub-haplogroup					
B4	12.9 (107)	16.4 (158)	14.8 (265)	0.76 (0.58–0.99)	0.039
B5	4.7 (39)	7.1 (69)	6.0 (108)	0.64 (0.43–0.96)	0.031
D4	10.1 (84)	10.9 (105)	10.5 (189)	0.94 (0.69–1.28)	0.676
D5	7.1 (59)	7.3 (70)	7.2 (129)	0.94 (0.65–1.36)	0.748
F1	14.0 (116)	8.6 (83)	11.1 (199)	1.72 (1.27–2.34)	0.001 *
F2	8.1 (67)	5.1 (49)	6.5 (116)	1.68 (1.13–2.48)	0.010
M7b	6.5 (54)	7.4 (71)	7.0 (125)	0.87 (0.60–1.27)	0.461
M7c	4.9 (41)	4.9 (47)	4.9 (88)	1.00 (0.64–1.55)	0.997

Asterisk denotes the significance after the Bonferroni’s correction (*: *p* < 0.0031).

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
