# Peer review of "Ischemic Stroke Risk Associated with Mitochondrial Haplogroup F in the Asian Population"

_cells, 2020, doi:10.3390/cells9081885_

Round 1

Reviewer 1 Report

The manuscript by Tsai et al has improved during revision, and relevant statistic comparisons are now more often (though not always) included to support the claims of the authors. If the below points can be remedied by the authors, I would judge the manuscript acceptable for publication in Cells:

-As previously requested, the authors now provide a HIF1alpha western blot in supplemental figure S1 with the key samples all run on the same gel. On Line 237, they state that the result is the same as in Fig 1 (where haplogroup B showed the highest HIF1alpha increase during hypoxia, and haplogroup F1 the lowest increase). However, this is not the case, as in Fig S1, haplogroup N9 shows the lowest HIF1alpha levels during hypoxia. Therefore, at the very least, the word “same” on line 237 should be exchanged for “similar”.

-Based on Fig 2C, LDH activity seems unchanged between normoxic and hypoxic conditions in all four tested haplogroups (no statistical significance is observed/marked on the plot). The interpretation on line 260-261 should therefore be adjusted to match the data.

-The description of the ROS measurements is somewhat confusing in its present form. The data show increased ROS upon hypoxia when measured in mitochondria (Fig 3a) and in the cytosol (Fig S2a). The mitochondrial ROS increases more in F1 than in B4, while the cytosolic ROS increases less or not at all in the F haplogroups than it does in the B haplogroups. However, on lines 288-292, the authors write: “Hypoxia increased intracellular ROS generation in both cybrids B and F, however cybrid B exhibited a notably higher level than cybrid F1 (Supplemental figure S2A). Conversely, we observed a marked mitochondrial ROS elevation in ischemic-susceptible mitochondrial haplogroup F1 (Figure 3A).”

For clarity, I would alter this statement for example to “Hypoxia increased mitochondrial and cytosolic ROS generation in both cybrids B and F (Figure 3A, Supplemental figure S2A).  However, while cybrid B exhibited a notably higher level of cytosolic ROS than cybrid F1 under hypoxic conditions (Supplemental figure S2A), the elevation in mitochondrial ROS was less pronounced in ischemic-susceptible mitochondrial haplogroup F1 than in cybrid B4 (Figure 3A).”

-I don’t agree with the interpretation of data in Figure 3C. On lines 309-311, the authors state: “We found that hypoxia-ischemia can activate entry of Ca2+ and increase cytosolic Ca2+ concentration in all cybrids (Figure 3C)”  although the cytosolic Ca2+ is only significantly increased in cybrid F2 and not in the others. Please modify the wording to match the data presented.

Reviewer 2 Report

The authors revised the manuscript significantly.   

Author Response

Comments and Suggestions for Authors

The authors revised the manuscript significantly.   

Submission Date

14 July 2020

Date of this review

29 Jul 2020 21:34:44

Response:

Thank you.

Reviewer 3 Report

The authors addressed most of issues pointed by the reviewers. 

Author Response

Comments and Suggestions for Authors

The authors addressed most of issues pointed by the reviewers. 

Submission Date

14 July 2020

Date of this review

23 Jul 2020 05:24:13

Response:

Thank you.

This manuscript is a resubmission of an earlier submission. The following is a list of the peer review reports and author responses from that submission.

Round 1

Reviewer 1 Report

This manuscript by Tsai et al. focuses on mitochondrial DNA haplogroups in connection to ischemic stroke risk and the associated hypoxia. They find that mtDNA haplogroup F, and specifically subgroup F1, is associated with increased risk of ischemic stroke in a group of 830 Taiwanese patients with a history of ischemic stroke compared to matched controls. They then generate cybrid cell lines containing mtDNA from various haplogroups and measure readouts of mitochondrial function in them under normoxia and hypoxia. Finally, they identify increased expression of angiopoietin-like 4 (ANGPLT4) in haplogroup F1 cybrids upon hypoxia; average ANGPLT4 protein levels are also increased in a smaller group of patients with acute ischemic stroke. This study therefore shows that haplogroup F1 is associated with a higher risk of ischemic stroke in the Taiwanese population, report some mitochondrial functions that could contribute to the effect, and identify ANGPLT4 as a promising candidate for future studies with regard to ischemic stroke.

The manuscript is generally clearly written, although in my opinion the flow of the results section could benefit from some explanation regarding the logic of the experiments, and some interpretation of the results (which is done very well in the discussion). With some exceptions, the conclusions are well supported by the data. I have some comments and suggestions regarding the presentation and discussion of the data that should be addressed or rebutted by the authors:

Broad comments:

- It is important that the procedure for quantification of the western blots is described in detail the Materials and Methods. It looks like the level of protein under normoxic conditions has been set to 1, and the level of protein upon hypoxia is presented as fold-change relative to that. Does that mean that the normoxic and hypoxic samples were run on the same blot? If not, how do the authors normalize signal to allow comparison of eg Hif1-alpha signal across blots? Signal intensity will depend on many factors such as antibody incubation, the length of exposure to the HRP substrate etc, which would not be corrected for by normalizing to the beta-actin loading control, since that is a different antibody and thus only controls for even loading. Similarly, it should be stated in the materials and methods or in the figure legend how the results of the other assays are treated - which is the control that each sample is compared to for each assay, for example in Fig 2D.

- Throughout the paper, the response of the various cybrid cell lines to hypoxia is compared. The most interesting aspect of these comparisons is often how one cybrid (eg. F1) responds compared to the other haplogroups, so whether F1 performs better or worse than the other cybrid lines. The statistical tests, however, are for the comparisons between the normoxic vs. the hypoxic condition of a specific cybrid line. For example, in Figure 2C, Rodamine 123 signal is shown as a read-out of mitochondrial membrane potential. Asterisks indicate whether the difference between normoxic and hypoxic conditions is significant for each cybrid line. However, in the text (line 232-233), it is stated: “mitochondrial membrane potential…was significantly decreased in haplogroup F1 compared to haplogroup B4 after hypoxia-ischemia”, although no significance testing has been carried out comparing B4 and F1. Similarly, for Fig 2D the authors write (lines 252-253): “markedly elevated levels of intra-mitochondrial calcium were found in F1 and F2 cybrids compared to haplogroup B cybrids”. There is no statistics presented that compares B cybrids to F1 or F2. In these and similar examples (Fig 3, Fig 5) throughout the manuscript, either statistics that supports the interpretation should be presented, or the statements should be rephrased.

- It is confusing that in some parts of the paper the authors discuss or imply that both F1 and F2 cybrids might be impaired in their response to hypoxia-ischemia (eg. line 335-337), while in other sections such as the RNA-seq analysis they focus only on genes that are up-regulated in F1 but not in F2. The clarity of the discussion would benefit from a clearer differentiation or interpretation of whether it is features found in both F haplogroups (the high ratio of mt vs. cytosolic ROS; higher increase in mitochondrial Ca2+, repression of PDH activity, higher induction of IL-1beta and IL-6; higher induction of MMP-6 but lower induction of MMP-2; ANGPLT4 protein level) or F1-specific ones (the low induction of HIF-1alpha; the high induction of ANGPLT4 RNA) that the authors think might be most relevant to ischemic stroke.

Specific comments:

- Line 76-77: Please specify which kit is used for genomic DNA isolation, and which polymerase is used for the multiplex pcr. Furthermore, oligo sequences should be presented (eg. in the supplemental data).

- Line 96: the info on the ELISA kit for ANGPTL4 appears to be wrong; RAB0204 is for GDF-15. Please correct.

- Line 103-111: Could the authors please describe, or provide a reference for, how the loss of mtDNA was verified in the rho0 cells and how the haplogroup of the generated cybrid cell lines was confirmed.

- Line 141: There is a typo: “camber” should read “chamber”.

-Line 150-151: Please include the manufacturer and ID for the beta-actin antibody.

- Line 156-157: There is an incomplete sentence, starting with “To measure absorbance…”.

- Line 164-165: For clarity, “was used to” should be “was by” or similar.

- In Fig 1, Fig 3, Fig 4 and Fig S1B,D, the length of the hypoxic treatment is not specified. Please add this to the figure or the figure legend.

- Line 251: What is meant by “store-operated” Ca2+ entry?

On the same line, the statement “increase(d) cytosolic Ca2+ concentration in all cybrids” is not supported by the data in Fig S1C; the increase is only significant in F1.

- Line 258-259, and 357: The authors write that the levels of inflammatory cytokines IL-1beta, IL6 an TNF-alpha are increased in haplogroup F cybrids. I don’t agree that the level of TNF-alpha can be stated to be increased in haplogroup F cybrids – it is only increased in F2, not in F1 (Fig 3A-C). Please correct.

- Line 323: It is claimed that the induction of HIF-1alpha is impeded in F1 and F2, although the data in Fig 1 only shows decreased induction in F1. Please correct.

Reviewer 2 Report

This study examines the role of mitochondrial genomic variations in ischemic stroke associated with atherosclerosis. The present study builds on the previous studies by other groups which showed that the risk of ischemic stroke is associated with different mtDNA haplogroups in different population. Here the authors report that ischemic stroke risk is associated with mitochondrial haplogroup F in Taiwanese population. The study demonstrated several cellular factors of vulnerability to ischemic stroke in the Mthapg F1 cybrids after hypoxia-ischemia which includes impeded induction of HIF-1a, decreased ATP production, lower mitochondrial membrane potential and higher mitochondrial ROS production. The results further showed an increase in the level of inflammatory cytokines was much higher in F1 cybrids than any other cybrids. Interestingly, the RNAseq study identified ANGPTL4 as a risk factor of stroke in Mthapg F1. ANGPTL4, the gene previously known to be associated with ischemic stroke, was elevated in F1 cybrids (but not any other cybrids) after hypoxic ischemia. Consistent with this, elevation of ANGPTL4 was found in the acute ischemic patient cohort compare to control.

This is an interesting study which demonstrates an evidence of relation between ischemic stroke risk and mitochondrial haplogroup F and identified ANGPTL1 as a risk factor of ischemic stroke associated with atherosclerosis in Taiwanese population.

There are, however, several issues that should be addressed by the authors:

Major points:

  1. Overall, there is a lack of rationale and interpretation in each subsection of results. It is understandable that the authors wrote rationale and interpretation in the Discussion, however, without them in the Results part, it is a kind of hard for readers to follow.

  1. According to the methods, authors stated a p-value of <0.0031 was considered statistically significant (this also needs to be stated in the first paragraph in the Result 3.1.1). The authors need to clarify if haplogroup B is significant or not, if not, it cannot be stated as below (see the underlined sentence). In addition, F2 is close to but not significant, so it needs to be clarified than just “marginal”. The whole paragraph is suggested to be improved.

Page 5, lines 194 – 202; Multiple logistic regression analysis revealed that haplogroup F was significantly associated with increased ischemic stroke risk; whereas haplogroup B was identified as significantly associated with decreased risk of ischemic stroke (OR 0.77, 95% CI=0.61-0.97, p=0.024). However, after Bonferroni correction, only haplogroup F remained significant (OR 1.44, 95% CI=1.14-1.82, p=0.002). Further sub-group analysis identified haplogroup F1 as significantly associated with increased risk of ischemic stroke (OR 1.72, 95% CI=1.27-2.34, p=0.001); meanwhile, although an association was also identified with haplogroup F2, this association was rendered only marginal after Bonferroni correction (OR 1.68, 95% CI=1.13-2.48, p=0.01)

  1. Along the same reason, in Table1, it shouldn’t be written “Asterisk denotes the significance at the 95% confidence level (*: p < 0.05)”, since p-value of <0.0031 is considered statistically significant. This may make readers confused. It would be better if the Table1 itself could be understandable which haplogroups are statistically significant.

  1. Since there is enough space in Figure2 and 3, it would be great to move Figure S1A-C to Figure 2. Also Figure S1D to Figure 3. They are important enough to be shown in the main figures, and in this way, the readers don’t need to be interrupted to go to another page to find the supplementary figures.

  1. The authors stated FAP has no known association with ischemic stroke. However, there are some literatures showed an association with stroke or inflammation. For examples, PMID: 27561653, PMID: 25464232, PMID: 21292680 etc. The authors need to describe in the Discussion at least.

  1. Though out the manuscript, authors compared F1 to B4 or B5. However statistically significance only show between normoxic and hypoxic condition in each cybrids. If authors state F1 is significantly higher/lower than B4/B5, appropriate statistics is needed.

Here is an example, in page 7, lines 231 – 233;

“In addition, mitochondrial membrane potential Δψm, which represents mitochondrial function, was significantly decreased in haplogroup F1 compared to haplogroup B4 after hypoxia-ischemia (Figure 2C).”

The authors stated mitochondrial membrane potential was significantly decreased in F1 compared to B4, but didn’t show the statistical significance between F1 and B4. Only showing significance between normoxic and hypoxic condition in each cybrids in the Figure 2D.

  1. Result 3.2.6; this paragraph needs to be improved with the rational and how the level of ANGPTL4 were measured. Result can be showed with a graph or table as well.

Minor points:

  1. In page 2, line 79 and line 87, “dichloride5” and “publication5”; the reference format needs to be fixed.
  2. In page 3, line 127, “75SE (Single-End or Paired-End)”; 75SE means it is prepared for single end. This sentence needs to be clarified.
  3. In page 4, lines 148 – 150, catalog number needs to be added for all antibodies.
  4. In page 4, line 154 and 158, “ATP 153 Colorimetric/Fluorometric Assay Kit (Catalog # K354-100)” and “Pyruvate dehydrogenase Enzyme Activity Microplate Assay Kit (ab109902)”; the vender needs to be added.
  5. In page 4, lines 169 – 170, “proper concentration” and “suitable time”; need to be clarified, what is working concentration and how long incubate.
  6. In page 4, lines 172 – 174, “The result was analyzed using the flow cytometry analysis software BD CellQuest (Becton Dickinson, San Jose, CA, USA). The data was analyzed using the flow cytometry analysis software WinMDI 2.8.”; it is unclear which software was used for analysis.
  7. In page 6, lines 206 – 207, “Hypoxia-inducible factor 1 alpha (HIF-1α) is a key factor regulating cellular adaptive and 206 survival responses to hypoxia-ischemia”; reference need to be added for this sentence.
  8. In Figure 2C, it seems normalized to normoxic condition in each cybrids, however, the value of normoxic condition for B5, F1 and F2 are not 100%. Please clarify it.
  9. In the figure legend for Figure 2, “(*p<0.05; **p<0.005; #p<0.0005)”; there are no **p and #p in this figure.
  10. Figure3C, according to the representative images and the graph, the level of TNF-alpha looks significantly increased in all four cybrids, however only B4 and F2 are showing significance between normoxic and ischemic condition. It needs to be clarified.
  11. In the figure legend for Figure 3, “(*p<0.05; **p<0.005; #p<0.0005)”; there are no **p and #p in this figure.

Reviewer 3 Report

The manuscript by Tsai et al, in which they investigated role of different mitochondrial haplogroup and possible mechanism of ischemic stroke. They identified in clinical subjects and further confirmed in culture model system. The manuscript is interesting, data well-presented and quite well written. However, there are few critical questions need to be addressed before publication. In various haplogroup, it is not clear about the role of various fuels like, glucose and fate of their downstream product like pyruvate. It would also be interesting to know pyruvate flux in PDH/LDH activation/inhibition situation and HIF1a/ROS level.

Author Response

Point-to-Point Response to the Reviewers’ Critiques

REVIEWER 3:

Comments and Suggestions for Authors

The manuscript by Tsai et al, in which they investigated role of different mitochondrial haplogroup and possible mechanism of ischemic stroke. They identified in clinical subjects and further confirmed in culture model system. The manuscript is interesting, data well-presented and quite well written. However, there are few critical questions need to be addressed before publication. In various haplogroup, it is not clear about the role of various fuels like, glucose and fate of their downstream product like pyruvate. It would also be interesting to know pyruvate flux in PDH/LDH activation/inhibition situation and HIF1a/ROS level.

Response:

I truly appreciate your insightful suggestion. I agree that clarifying the pyruvate flux in PDH/LDH activation/inhibition, and the HIF1a/ROS level would be interesting and provide further insight into the pathogenesis of mtDNA haplogroup and the association with ischemic stroke. However, due to time constraints of the revision process this experiment will have to be part of our future investigations.

Round 2

Reviewer 2 Report

The revision addressed most of the reviewers' points properly. 

However, a further change is recommended as below;

In Table 1, since the authors state only haplogroup F is significant, it is suggested to delete "*: p < 0.05". Therefore, the sentence "Asterisk denotes the significance at the 95% confidence level" also needs to be changed pertinently.

Reviewer 3 Report

not recommended